# Mesenchymal Characteristics and Predictive Biomarkers on Circulating Tumor Cells for Therapeutic Strategy

**DOI:** 10.3390/cancers12123588

**Published:** 2020-11-30

**Authors:** Takahiro Okabe, Shinsaku Togo, Yuichi Fujimoto, Junko Watanabe, Issei Sumiyoshi, Akira Orimo, Kazuhisa Takahashi

**Affiliations:** 1Leading Center for the Development and Research of Cancer Medicine, Juntendo University Graduate School of Medicine, 2-1-1 Hongo, Bunkyo-ku, Tokyo 113-8421, Japan; t-okabe@juntendo.ac.jp; 2Division of Respiratory Medicine, Juntendo University Faculty of Medicine & Graduate School of Medicine, 2-1-1 Hongo, Bunkyo-ku, Tokyo 113-8421, Japan; yfujimo@juntendo.ac.jp (Y.F.); ju-watanabe@juntendo.ac.jp (J.W.); i-sumi@juntendo.ac.jp (I.S.); kztakaha@juntendo.ac.jp (K.T.); 3Research Institute for Diseases of Old Ages, Juntendo University Graduate School of Medicine, 2-1-1 Hongo, Bunkyo-ku, Tokyo 113-8421, Japan; 4Departments of Pathology and Oncology, Juntendo University School of Medicine, 2-1-1 Hongo, Bunkyo-ku, Tokyo 113-8421, Japan; aorimo@juntendo.ac.jp

**Keywords:** circulating tumor cells, circulating tumor microemboli, epithelial–mesenchymal transition

## Abstract

**Simple Summary:**

Circulating tumor cells (CTCs) are the seeds that spread through the circulatory system and generate the metastatic sites. Among the various phenotypes exhibited by CTCs, epithelial-to-mesenchymal transition (EMT) has extensively gained attention because of its contribution to the acquisition of invasiveness and motility of CTCs, which is critical for the successful metastasis process. So far, many clinical studies have demonstrated that expressions of mesenchymal features in CTCs are associated with poor clinical outcomes in many types of solid cancer. In this review, the clinical significance of CTCs as a liquid biopsy is described in terms of the mesenchymal characteristics, expression of predictive biomarkers, and their correlations. Detailed analysis of these multivariate targets in CTCs could improve therapeutic decision-making.

**Abstract:**

Metastasis-related events are the primary cause of cancer-related deaths, and circulating tumor cells (CTCs) have a pivotal role in metastatic relapse. CTCs include a variety of subtypes with different functional characteristics. Interestingly, the epithelial–mesenchymal transition (EMT) markers expressed in CTCs are strongly associated with poor clinical outcome and related to the acquisition of circulating tumor stem cell (CTSC) features. Recent studies have revealed the existence of CTC clusters, also called circulating tumor microemboli (CTM), which have a high metastatic potential. In this review, we present current opinions regarding the clinical significance of CTCs and CTM with a mesenchymal phenotype as clinical surrogate markers, and we summarize the therapeutic strategy according to phenotype characterization of CTCs in various types of cancers for future precision medicine.

## 1. Introduction

The major cause of death in ~80% of cancer cases is a metastatic cancer lesion-related complication. Increasing numbers of circulating tumor cells (CTCs) are now utilized as surrogate endpoint markers [1,2,3,4,5], and many clinical studies have demonstrated that the presence of CTCs in the blood of patients associate with short recurrence-free survival and high risk of metastasis for several types of solid cancers [6,7,8,9,10,11,12,13]. Thus, CTCs are a glaringly obvious direct seed for forming future micro-metastatic colonies in distal organs. However, micro-metastatic colonization is a challenging process as the minorities of the surviving cells are exposed to significant shear force, innate immunity, chemotherapy, and oxidative stress under anoikis (detachment-induced apoptosis) in the bloodstream. In this respect, epithelial-to-mesenchymal transition (EMT) is related to survival by allowing cancer cells to avoid apoptosis and acquire the subsequent chemoresistance even under anoikis in the bloodstream [14]. The monitoring of CTCs could lead to the prediction of the future clinical time course of cancer development and/or early recurrence period at the onset of initial cancer development and to the detection of micro-metastatic lesions [15]. Therefore, the significance of CTCs as clinical surrogate markers is different from that of other diverse liquid biopsies (i.e., circulating tumor DNA, ctDNA, exosomes, or diverse serum liquid biomarkers) [16,17,18,19]. Among these, ctDNA measurement is the most popular diagnostic and prognostic tool that averages the status of the whole tumor burden and provides information on the response/resistance to chemotherapy, targeted therapy, and immunotherapy; therefore, it is useful for decision-making [17,18]. However, these biomarkers are considered as indirect markers released from tumor tissues into the bloodstream. In contrast, the detection of CTCs and circulating tumor microemboli (CTM) provide direct evidence for the existence of cancer lesions, and thus their measurement and phenotypic characterization could be a reliable diagnostic tool for cancer. The mesenchymal features of CTCs have been reported and, based on the expression level of epithelial and/or mesenchymal markers in CTCs, three different CTC subtypes—epithelial CTCs, hybrid CTCs that involve both epithelial and mesenchymal markers, and mesenchymal CTCs—were identified [20] (Figure 1). CellSearch^®^ is the most commonly used detection method for CTCs worldwide, utilizing antibodies against the epithelial markers epithelial cell adhesion molecule (EpCAM) and cytokeratin. CTC detection by the CellSearch^®^ system (generally two or more CTCs/7.5 mL blood as positive) does not have a high sensitivity in non-small cell lung cancer (NSCLC) and pancreatic ductal adenocarcinoma (PDAC), giving a result of only 20% even in the advanced metastatic stage [21]. This contributes to the underestimation of CTC number and could be, in part, due to the high number of CTCs with a mesenchymal phenotype undergoing EMT (i.e., from hybrid CTCs to EMT-CTCs) [22,23]. Similarly, a study conducted on malignant pleural mesothelioma using the CellSearch^®^ system showed a low detection rate of 33%, suggesting a need for efficient EpCAM-independent detection systems of CTCs [24]. CTCs morphologically and biologically involve different phenotypes and consist of single cells and/or cell clusters with or without the EMT feature, which plays an essential role in the metastatic process. CTM is also seen in patients with advanced cancers [25] and is detected as multicellular cancer fragments, ranging from 2 to 100 cancer cells (with most clusters comprising between 20 and 40 cells), in the blood circulation of patients with diverse metastatic cancers [25,26,27,28,29,30]. Cluster CTCs also contain cancer-associated fibroblasts, platelets, and immune cells [31]. CTM appears to be more aggressive than single CTCs in developing cancer invasion [32,33,34]. However, it remains unclear which types of CTCs (single versus clusters) play a critical role in the metastasis process. 

In the past decade, the detection of CTCs as clinical biomarkers has been reported using diverse, highly sensitive methods. A randomized trial including patients with metastatic breast cancer with a persistent increase in CTCs reported that an early switch to an alternative cytotoxic therapy after the failure of the first-line chemotherapy prolonged the overall survival (OS) (SWOG S0500) [5]. Recent CTC-related studies focused on further detailed approaches, including phenotypic characterization of predictable therapeutic response markers through genotypic and immunophenotypic analysis. These approaches can provide useful predictive information for the decision-making in precision medicine, but CTCs represent a heterogeneous population of tumor cells with the potential for metastases [35]. Thus, CTC measurements to predict treatment efficacy are still insufficient in providing useful clinical biomarkers with clear evidence. 

Here, we summarize the clinical significance of CTCs, including the mesenchymal phenotype and the potential treatment response markers for future precision medicine.

## 2. The Significance of EMT in CTCs as Prognostic and Predictive Biomarkers

It is widely known that EMT contributes to the generation of CTCs by facilitating intravasation into the blood circulation [36]. EMT is induced by transcriptional factors, and the Twist–Snail axis plays a vital role in the transcriptional regulation of EMT and metastasis to maintain the stability and survival of CTCs in the bloodstream by avoiding anoikis suppression [37]. Zeb1, which acts as a repressor for E-cadherin transcription, protects anoikis-induced apoptosis and causes metastatic tumors in vivo [38]. Thus, these transcriptional regulators of EMT are also utilized as detection markers for EMT-CTCs. Vimentin, a cytoplasmic protein comprising intermediate filaments, is known to be upregulated in EMT-induced cancer cells and correlated with cancer progression [39,40,41]. In addition, some vimentin molecules expressed in cancer cells migrate to the cell surface and have been identified as cancer-specific EMT markers, called cell-surface vimentin (CSV), and were shown to correlate with highly invasive and metastatic properties of cancers [42,43,44,45,46,47]. Subsequent mesenchymal–epithelial transition (MET), the reverse process of EMT, in the metastatic cascade for extravasation into secondary metastatic sites [48] reportedly induces the re-expression of E-cadherin [49]. EMT leads not only to recurrent metastasis but also to chemoresistance. Yu et al. followed the epithelial/mesenchymal marker status of hybrid CTCs. When patients with breast cancer responded to chemotherapy, epithelial marker-dominant CTCs were dramatically reduced, and when patients developed recurrence, the amount of mesenchymal marker-dominant CTCs was significantly increased [26]. Furthermore, blocking EMT contributed to enhanced chemosensitivity and increased the OS of genetically engineered cancer mouse models [50]. Patients with advanced gastric cancer and breast cancer showed an increase in mesenchymal CTCs in the post-chemotherapy blood specimens, which was associated with the progressive disease, suggesting that monitoring the EMT status of CTCs can provide reliable information to predict the therapy response and cancer recurrence for future precision medicine [22,26,51,52]. In a study conducted for primary breast cancer, 77 (18.0%) of 427 primary breast cancer patients were found to have EMT-CTCs characterized by TWIST, SNAIL1, SLUG, and ZEB1 expression levels, and patients without EMT-CTCs had a longer disease-free survival than those with EMT-CTCs [53]. Positive EMT-CTCs at the baseline are associated with poor clinical outcome for treatment responsiveness in patients with advanced NSCLC [23]. The presence of mesenchymal markers, Twist1+, vimentin+, or CSV+, in CTCs predicts a worse prognosis than the expression of epithelial markers alone in metastatic cancers [46,47,54]. A study of metastatic breast cancer has demonstrated that overexpression of EMT-inducing transcription factors is associated with shorter progression-free survival (PFS), and EMT-CTCs are associated with recurrence [55]. Furthermore, patients with metastatic diseases have a high incidence of EMT-CTCs compared to those with early-stage diseases [56,57,58,59]. In contrast, EMT-CTCs have been reported as dispensable biomarkers to predict prognosis in patients with castrate-resistant prostate cancer (CRPC), PDAC, and breast cancer [60,61,62]. Furthermore, inhibiting EMT by overexpressing the microRNA miR-200 does not affect lung metastasis but abrogates EMT-mediated chemoresistance [63]. Patients with PDAC show KRAS mutations in CTCs [64,65], indicating a prognostic value, which does not depend on the expression of ZEB1, the EMT marker [66]. These negative results for the value of EMT-CTCs as prognostic markers are probably due to the diverse mechanisms that induce EMT and the lack of ability to detect EMT-CTC subtypes in clinical settings [67]. The clinical significances of CTCs with mesenchymal phenotype are summarized in Table 1.

Since mesenchymal markers such as N-cadherin and vimentin are frequently expressed in blood cells, it is necessary to remove these blood cells during the enrichment and detection processes of CTCs. One approach to enable blood cell removal is the negative selection with blood cell-specific markers, such as CD45, not expressed in cancer cells. Although there are no universal markers for specific enrichment and detection of mesenchymal CTCs, various detection systems have combined negative selection of blood cells with positive selection by mesenchymal markers to achieve highly sensitive EMT-CTC detection [85,86,87,88,89].

### 2.1. The Early Occurrence of EMT in CTCs

A large meta-analysis of 19 studies enrolling 2993 early-stage breast cancer patients demonstrated the detection of the CTCs as a stable prognosticator in early-stage breast cancer [90]. Several studies have identified CTCs expressing EMT markers in patients with early-stage breast cancer [57,91,92]. Mego et al. reported that CTCs exhibiting EMT phenotype are more frequent than those expressing only epithelial markers [91]. Furthermore, RT-PCR characterization of CTCs isolated from early-stage breast cancer patients, where Twist1 expression was used as an EMT marker, revealed the presence of EMT-CTCs in 31% of patients [93], whereas immunostaining experiments revealed that EMT-CTC expressing Twist1 and vimentin were identified in 73% and 77% of patients with early-stage breast cancer, respectively [57]. The occult micro-metastatic tumor cells establish cytokeratin-positive cells in the lymph nodes, and the overexpression of EMT markers, occurring at an early stage, is associated with a worse OS even in patients with curatively resected stage I NSCLC [94,95,96]. CTC-positive patients with NSCLC and breast cancer at the time of definitive surgery showed short recurrence-free periods [97,98,99]. Thus, CTCs isolated from early stages of lung cancer are predictors of poor prognosis and can be used to determine biomarkers predictive of early recurrence [35]. EMT-CTCs were identified in patients with early-stage NSCLC before receiving surgical operation, and the population of EMT-CTCs was recognized at an almost 50% lower rate than that in patients with some other solid tumors [23,68]. In a previous study, CTCs with EMT markers were detected in a high-risk lung cancer group without clinically detectable tumors before the occurrence of early-stage lung cancer [100]. Thus, the development of a system with EMT-CTC detection capability will enable high-sensitivity CTC detection from early stages of cancer to advanced metastatic stages compared to systems that use only epithelial markers to detect CTCs such as the CellSearch^®^ system.

### 2.2. The Metastatic Ability of CTCs and CTC Clusters with a Mesenchymal Phenotype

The increase in the percentage of CTCs positive for the mesenchymal marker is higher in metastatic breast cancer than in early-stage breast cancer [57], denoting a more aggressive and metastatic potential of mesenchymal CTCs than those with epithelial markers alone [56,73,76,78,79,80,101,102]. Hybrid CTCs, cytokeratins^+^/vimentin^+^/CD45^−^, are also associated with disease progression and the presence of distant metastasis [103,104]. EMT-CTCs have specific markers that allow a more accurate prediction of worse prognosis and metastasis. BCAT1-induced EMT-CTCs may be an important biomarker of hepatocellular carcinoma metastasis [105]. Twist+ CTCs with a higher expression of glypican-3 are closely correlated with the rate of metastasis, recurrence, and mortality [105]. EMT-CTCs with high LGR5 expression significantly correlate with the occurrence of metastasis [105]. These findings suggest that EMT-CTCs play a crucial role in developing initial metastatic colonization at distal organs. A model for cancer metastasis has been proposed in which EMT promotes the release of both EMT and non-EMT cells from the primary tumor and vascular invasion, and non-EMT cancer cells in the bloodstream collaborate with EMT cells to establish colonies in the secondary sites [106]. CTC clusters correlate with higher metastatic potential and shorter survival times than single CTCs [107,108,109,110,111]. CTC clusters increase the metastatic potential by approximately 50-fold compared with single CTCs in experimental models [33]. CTM maintains the epithelial feature of preserving cell–cell junctions, and CTM displaying plakoglobin, a cell adhesion protein that contributes to the CTC cluster formation, is an independent prognostic factor, particularly for distant-metastasis-free survival in patients with breast and prostate cancer [33]. Platelets bound to CTC clusters in the blood circulation release TGF-β, which leads to EMT in the CTC cluster, and this partial EMT status in the hybrid CTC cluster may have a higher metastatic potential and a survival advantage during colonization at the distant site [26,112]. 

## 3. Cancer Stem Cells (CSCs) with a Mesenchymal Phenotype

CTCs expressing CSC markers indicate tumor recurrence and exhibit significant resistance to conventional chemotherapy [113]. EMT promotes the acquisition of CSC-like properties [114], and the CSC phenotype also expresses genes associated with the EMT features, resulting in enhanced metastatic ability [115]. Cases of gastric cancer with CD44^+^ in CK^+^ epithelial CTCs are significantly more common among patients with distant metastases and associated with shorter survival than that observed in the case of patients with CK^+^CD44^−^ CTCs [71]. NSCLC patients with an increased ratio of CD133^+^ CTCs to pan-CK-positive cell type (stem cell-like to epithelial ratio) and those that have mesenchymal N-cadherin-positive CTCs are significantly associated with shortened PFS [116]. The co-expression of the stem cell marker CD133 in hybrid CTCs (CK^+^, vimentin^+^, and N-cadherin^+^) has also been detected at a high frequency in metastatic breast cancer [117,118]. Aldehyde dehydrogenase 1 (ALDH1), a CSC marker, has potential diagnostic and therapeutic implications [119], and ALDH1+ CTCs positivity in patients with NSCLC was found to be associated with the disease stage [120]. Additionally, ALDH1^+^ EMT-CTCs (vimentin^+^ and fibronectin^+^) are isolated from patients at a higher frequency with metastatic breast cancer and correlated with the disease stage [121]. The expression of both ALDH1 (ALDH 1^high^) and nuclear Twist (Twist^nuc^) CTCs in patients with metastatic breast cancer is higher than that in patients with early-stage breast cancer [92]. CTCs with positive EMT markers and/or ALDH1 expression indicate therapy resistance in patients with metastatic breast cancer [122]. The expression of the stemness markers CD44^+^/CD24^−^ and ALDH1^+^ in CTCs is associated with an enhanced tumorigenic potential in breast cancer [123]. The glioblastoma-derived CTC expressing the CSC phenotype, which includes the stemness-associated genes SOX2, OCT4, and NANOG, contributes to local tumorigenesis and recurrence by the process of self-seeding [124]. These studies indicate that the CTCs that acquire CSC properties, namely circulating cancer stem cells (CTSCs), are more frequent in advanced-stage cancers and are associated with poor clinical outcomes. The clinical significances of CTSCs are summarized in Table 2.

## 4. The Concordance between the Characteristics of CTCs and Primary Tumors

Detection of CTC driver mutation as a predictable molecular targeted therapy response, which reflects the concordance with primary tissue status, is essential for non-invasive therapeutic decision-making. The concordance of KRAS status, a biomarker for predicting the therapeutic efficiency of anti-epidermal growth factor receptor (EGFR) therapy, between CTCs and the corresponding colorectal cancer tumor tissues was 77% [125]. Similarly, Buim et al. also showed that KRAS mutations (codon 12 and 13) in CTCs and matched primary tumor from patients with colorectal cancer showed a high grade of concordance of 71% (*p* = 0.017), which suggests that CTCs can be used as surrogates of primary tumors in clinical practice [126]. Several other studies have also shown similar results [126,127,128,129]. Kalikaki et al. characterized CTCs from 31 metastatic colorectal cancer patients for KRAS mutations on codons 12 and 13 and showed 45% and 16.7% of patients with mutant and wild-type primary tumors, respectively, have detectable mutations in their CTCs [130]. They further revealed that individual patients’ CTCs in serial blood samples exhibited different mutational statuses of KRAS during treatment [130]. Furthermore, it has been shown that the frequency of at least one matching KRAS mutation between CTCs and PDAC was 58% [64], and the discordant KRAS mutations in CTCs and corresponding tumors were not associated with OS [64,66]. These studies show the importance of utilizing CTCs as a dynamic source of liquid biopsy for real-time genotyping of tumor cells during treatment. 

In breast cancer, the human epidermal growth factor receptor 2 (HER2) oncogene has been shown to play an important role in growth and progression [131,132]. A good correspondence of HER2 amplification was found between CTCs and the primary tumor as well as the immunohistochemistry results [133], suggesting that the status of HER2 on CTCs reflects the status of the primary tumor. However, several studies have demonstrated that HER2 status might change during disease progression [134,135,136,137]. Moreover, discordance between the expression of HER2 and the estrogen receptor (ER) in CTCs and that in breast cancer tumor tissues occurs frequently [138], and thus has no prognostic impact on patients with metastatic breast cancer with a HER2-negative primary tumor receiving endocrine therapy [134,138,139]. These discordances could be, in part, due to the inability of traditional biopsies that sample a small piece of tumor tissue to reflect the heterogeneity of cancer cells in tumor tissue leading to failure in cancer prognosis and prediction. In contrast, several studies have provided evidence that the heterogeneity of CTCs reflects the cellular heterogeneity of the primary and metastatic tumor [140,141,142]; moreover, CTCs often acquire different characteristics from their original tissue during the metastatic process. Therefore, a detailed characterization of the expression of CTC markers could provide useful information in predicting prognosis and treatment response as a real-time biopsy [139,143].

## 5. The Potential Markers on CTCs for Precision Medicine

Detection of predictive biomarkers for treatment response on CTCs can improve clinical decision-making, and many studies have been conducted to identify reliable biomarkers. These potential treatment response markers for future precision medicine are summarized in Table 3.

### 5.1. The Analysis of Molecular Target Expression in CTCs as a Predictive Biomarker for Treatment Response

HER2 gene amplification in CTCs was determined during cancer progression, even in patients with HER2-negative primary tumors [144]. Detected HER2-positive CTCs in patients with early-stage breast cancer [145,146] and patients that had HER2-positive CTCs before neoadjuvant therapy were associated with shortened disease-free survival and OS [35]. The predictive treatment response values of HER2-positive CTC status have been demonstrated in patients receiving anti-HER2 therapy [147]. However, trastuzumab, an anti-HER2 antibody therapy, did not decrease the detection rate of CTCs in patients with non-HER2-amplified early breast cancer following surgery after chemotherapy [148]. A digital RNA analysis of CTCs demonstrated that the failure to suppress ER signaling within CTCs after endocrine therapy predicts early progression, and therefore, scoring the CTC-RNA signatures could guide the therapeutic options [149]. CTCs with synaptophysin expression are also a useful biomarker of both the treatment efficacy of and resistance to androgen receptor-targeted therapies in patients with metastatic CRPC [150]. Other specific and predictable chemoresponse markers in CTCs have also been reported. In patients with metastatic breast cancer, a decrease in the topoisomerase 1 (Top1) level in CTCs after the initial cycle of administration of the Top1 inhibitor etirinotecan pegol was associated with an increased OS [151]. During treatment, Bcl-2^+^/CD45^−^ CTCs seemed to be a dynamic biomarker for predicting the treatment efficacy and clinical outcome in patients with chemotherapy-naive small cell lung cancer (SCLC) [152].

### 5.2. The Analysis of Oncogenic Driver Mutation in CTCs as a Predictive Biomarker for Treatment Response

The efficacy and prognosis of EGFR-tyrosine kinase inhibitor (TKI) treatment for patients with EGFR driver mutations were reportedly better in the low CTC group than in the high CTC group in patients with NSCLC [153]. However, CTCs contain multiple EGFR mutations due to CTC heterogeneity, as evidenced by next-generation sequencing [154]. In patients with EGFR-mutant tumors progressing on EGFR-TKI therapy, 70% of CTCs contained the EGFR T790M mutation and acquired EGFR-TKI resistance [153]. Anaplastic lymphoma kinase (ALK) rearrangement with aberrant ALK-fluorescence in situ hybridization (FISH) patterns can be detected in CTCs in concordance with patients having ALK-positive NSCLC, enabling both diagnostic testing and decision-making for ALK inhibitors. These ALK-rearranged CTCs also express a mesenchymal phenotype and decrease the response to crizotinib [155]. Decreasing ALK copy number in CTCs is associated with treatment response and a longer PFS [156]. Single-cell sequencing of CTCs from ALK-rearranged patients may provide information to assess mutations that cause therapeutic resistance to ALK inhibitors [157]. Androgen receptor (AR) amplification is conserved between CTCs and tumor biopsies in patients with metastatic CRPC, and CTCs can serve as a non-invasive surrogate to document AR amplification [158]. It has been demonstrated that AR splice variant 7 (AR-V7) expression in CTCs and CTC clusters predicts poor treatment response in metastatic CRPC patients treated with novel hormonal therapies, such as abiraterone or enzalutamide [159,160]. CTCs expressing AR-V7 and ARv567es+, as well as the percentage of androgen receptor nuclear localization, are associated with sensitivity to taxane therapy [161,162]. Phenotypic profiling of CTCs isolated from patients with metastatic CRPC (mCRPC) has revealed elevated levels of AR-V7 and the mesenchymal marker, N-cadherin, in patients with progressive cancer [163].

### 5.3. The Immunophenotype Analysis of CTCs as Predictive Biomarkers for Immune Checkpoint Therapy Response

Programmed death-ligand 1 (PD-L1) is widely implicated in tumor immune evasion, and PD-L1 expression in tumor tissues has been reported to be associated with the responses to the current immune checkpoint therapy of melanoma and NSCLC [164,165,166,167,168]. Therefore, PD-L1 expression in CTCs is used as a predictive response value for PD-1/PD-L1 inhibitor immune checkpoint therapies. However, the concordance of PD-L1 expression between tumor tissues and CTCs is not high [169,170,171]. This might support the conclusion that the presence of PD-L1^+^CTC is not significantly correlated with the treatment outcomes of nivolumab, a PD-1 inhibitor, and a higher baseline PD-L1^+^CTC number does not lead to a response in patients with advanced NSCLC [172]. Interestingly, the levels of EMT-CTCs are negatively correlated with T-cell immunity, and the levels of CD3^+^ and CD8^+^ T cells in NSCLC suggest resistance to cancer immunity [173]. Patients who show an increase in PD-L1^+^CTCs are predicted to have resistance to PD-1/PD-L1 inhibitors, while no changes or a decrease in PD-L1^+^CTCs is observed in responding patients [170]. Patients with advanced melanoma and gastrointestinal tumors with detectable PD-L1^+^CTCs also responded to the PD-1 inhibitor and longer PFS [174,175]. A higher number of CSV^+^PD-L1^+^ CTCs are significantly associated with short survival and poor therapeutic responses in gastric cancer patients [46]. 

### 5.4. The Combination Analysis of EMT and Predictive Markers for Treatment Decision-Making

The existence of EMT-CTCs was found to correlate with drug resistance [54,184]. Therefore, the detection and monitoring of EMT as a marker of treatment resistance before and during treatment is important to improve the quality of treatment management. High EMT marker expression has also been associated with high expression of immune checkpoint markers, such as PD1, PD-L1, CTLA4, OX40L, and PD-L2 [185,186,187,188]. For example, several studies have shown that EMT in cancer cells increases the expression of PD-L1 and consequently confers immune tolerance in cancer cells [72,189,190,191,192]. Analyses of CTCs in NSCLC patients revealed that PD-L1 expression on EMT-CTCs associated with significantly poorer survival after curative surgery, and PD-L1 and EMT markers were expressed at significantly higher proportions in CTCs than patient-matched NSCLC tissues [72,190]. Furthermore, it was reported that PD-L1 expressed in the nucleus of vimentin-positive CTCs predicts poor prognosis in colorectal and prostate cancer patients [193]. Several studies suggest complex bidirectional regulation between EMT and PD-L1 signaling, leading to tumor immune escape [194,195,196]. The combined analysis of EMT and PD-L1 expression in CTCs could be an effective method for predicting and monitoring the outcome of immunotherapy. EMT-related genes in CTCs are also candidate markers for combination analysis. For example, in metastatic prostate cancer, IGF1, IGF2, EGFR, FOXP3, and TGFB3 were commonly observed in CTCs of castration-sensitive cases, whereas an additional subset of EMT-related genes (e.g., PTPRN2, ALDH1, ESR2, and WNT5A) were expressed in CTCs of castration-resistant cases [197]. Mesenchymal phenotype and increased expression level of AR-V7 in CTCs from patients with mCRPC have been observed for those with a limited response to abiraterone and enzalutamide (androgen-signaling targeted inhibitors) treatment [163]. Further detailed molecular analysis of the relationship between EMT and predictive biomarkers could lead to a better understanding of the mechanisms of treatment resistance and create a reliable basis for therapeutic decision-making.

## 6. Conclusions

The detailed molecular characteristics, including mesenchymal and cancer stem cell features, in single CTC and CTM are required to develop reliable and predictable biomarkers as prognostic, diagnostic, and therapeutic strategies. However, CTCs have a diverse and heterogeneous phenotype, and thus, the clinical value of CTCs and CTM with mesenchymal phenotype still does not translate into clinical applications. Furthermore, the expression of some pharmaco-molecular targets in CTCs is different from that in primary tumors, making it difficult to standardize the role of CTCs as a companion diagnostic tool. On the contrary, the CTCs and CTM with a mesenchymal phenotype are associated with poor clinical outcomes and provide information on recurrence periods and therapeutic resistance, rendering them as a potential source of predictable biomarkers. Therefore, detailed analyses of multivariate targets in CTCs could improve clinical decision-making as a real-time liquid biopsy. Furthermore, molecular studies of the relationship between EMT and immune checkpoint will lead to new cancer immunotherapies based on EMT markers and immune checkpoint status. This study suggests that, for the development of specific and reliable therapeutic markers based on CTCs and CTM, further clinical prospective studies should be undertaken to increase the clinical value of CTC and CTM measurements for future personalized treatments. 

## Figures and Tables

**Figure 1 cancers-12-03588-f001:**
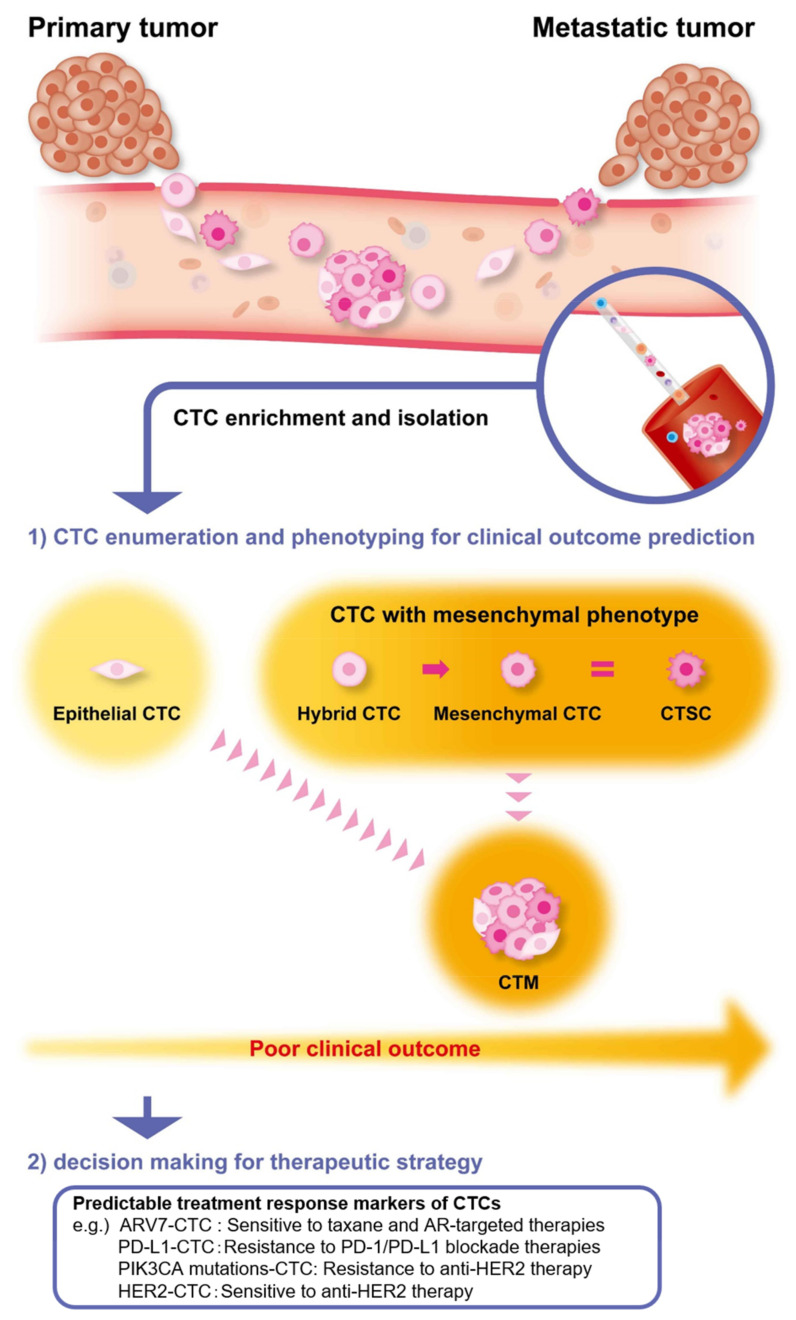
Schematic representation of circulating tumor cells (CTCs) during the metastatic process and characterization as a clinical biomarker after isolation. During the metastatic process, CTCs undergo gradual or full epithelial-to-mesenchymal transition (EMT) by which the primary epithelial tumor cells acquire mesenchymal characteristics and enhance metastatic and invasive abilities. Circulating tumor stem cells (CTSCs) exhibit the stem cell properties and the mesenchymal phenotype. Circulating tumor microemboli (CTM) are the CTC clusters, which often appear to be a heterogeneous cell population. The presence of CTCs and CTM with mesenchymal markers associate with chemoresistance and poor clinical outcomes, and the detection of these cells has significance as surrogate biomarkers for decision-making in therapeutic strategy.

**Table 1 cancers-12-03588-t001:** Mesenchymal markers expressed on circulating tumor cells (CTCs) from patients with different tumor types and clinical significance as surrogate biomarkers.

Cancer Type	Molecular Marker on CTC	Detection Method	Significance as Biomarker	Positivity of Total Pts	Ref.
Advanced NSCLC	VIM	CellSearch^®^	Increasing in the EGFR-mutated subgroup	20.80%	[68]
Gastric cancer	VIM	Cyttel method	Immune evasion capacity of CTCs	75.86%	[69]
Gastric cancer	CSV/PD-L1	CSV magnetic positive selection	Short survival duration and poor therapeutic response	71%	[46]
Lung cancer	VIM	Vimentin-iFISH	Shorter PFS	19.70%	[70]
Gastric cancer	CK+/CD44+	Flow cytometry sorting	Shortened survival	10%	[71]
NSCLC	VIM/N-Cadherin/PD-L1	Immunofluorescent staining	Poorer survival after curative surgery	86.70%	[72]
NSCLC	VIM/Twist	CanPatrol™	Predictor of metastasis	80.00%	[73]
mBC	ALDH1/TWIST1	Immunofluorescent staining	Chemoresistance, lung metastases, and decreased PFS	27.70%	[22]
CRPC	EpCAM^low-expression^	CellSearch^®^	Shorter OS in ≥5 EpCAM^high-expression^ CTC	28.00%	[74]
mBC	EpCAM^low-expression^	CellSearch^®^	Shorter OS in ≥5 EpCAM^high-expression^ CTC	36%	[74]
Colorectal cancer	LGR5+	CanPatrol™	Metastasis	86.40%	[75]
mBC (triple negative subtype)	VIM	Microfluidic Chip device	Shorter PFS	100%	[51]
Esophageal squamous cell carcinoma	VIM/Twist	CanPatrol™	Staging	32.60%	[76]
Hepatocellular carcinoma	VIM/Twist	CanPatrol™	Shortened postoperative disease-free survival	38.79%	[75]
Breast cancer	TUB/VIM/GLU	ISET	Metastases	8.24% (VIM)	[77]
Hepatocellular carcinoma	VIM/Twist	CanPatrol™	Metastasis	90.18%	[78]
Colorectal cancer	AKT2/SNAIL1	CanPatrol™	Staging and metastasis	56.90%	[79]
Colorectal cancer	VIM	CanPatrol™	Metastasis	42.06%	[80]
Hepatocellular carcinoma	VIM/Twist	CanPatrol™	Shortened postoperative disease-free survival	69.20%	[81]
Prostate cancer	CK/VIM	Parsortix system	Tumor aggressiveness and poorer survival	73%	[59]
NSCLC	VIM	TelomeScan^®^	Shorter PFS	46%	[23]
Ovarian cancer	PI3Kα/Akt-2/Twist	AdnaTest OvarianCancer and EMT-1	Suggested therapeutic resistance	30%	[82]
mBC	VIM/Twist	CanPatrol™	Increasing in the negative hormone receptor subgroup	93%	[83]
mCRPC	VIM	CellSearch^®^	Shorter OS	32.30%	[84]

VIM: vimentin; CSV: cell-surface vimentin; CK: cytokeratin; mCRPC: metastatic castration-resistant prostate cancer; NSCLC: non-small cell lung cancer; SCLC: small cell lung cancer; mBC: metastatic breast cancer; Pts: patients.

**Table 2 cancers-12-03588-t002:** Circulating cancer stem cell (CTSC) markers from different tumor types and clinical significance as surrogate biomarkers.

Cancer Type	Molecular Marker on CTC	Detection Method	Clinical Significance as Biomarker	Positivity of Total Pts	Ref.
Gastric cancer	CK^+^/CD44^+^	Flow cytometry sorting	Shortened survival	10%	[71]
NSCLC	CD133	Immunofluorescent staining	Shortened survival	84%	[116]
CRPC	CD133	CellSearch^®^	Metastases	82%	[117]
Breast cancer	ALDH1	Immunomagnetic cell selection	Recurrence and survival	14%	[118]
NSCLC	ALDH1	RT-PCR following immunomagnetic cell selection	Progressive metastases	25%	[120]
mBC	ALDH1^+^/VIM^+^/fibronectin^+^	Immunomagnetic cell selection	Metastases	34%	[121]
mBC	ALDH1/nuclear Twist	Immunofluorescent staining	Disease progression	76%	[92]
mBC	EMT markers (TWIST, Akt2, PI3K) and/or ALDH1	Immunomagnetic cell selection	Therapy resistant	38%	[122]
Breast cancer	CD44^+^/CD24^+^ ^or^ ^low-expression^	Immunofluorescent staining	Metastases	23%	[123]

CK: cytokeratin; ALDH1: aldehyde dehydrogenase-1; VIM: vimentin; NSCLC: non-small cell lung cancer; CRPC: castration-resistant prostate cancer; PFS: progression-free survival; EMT: epithelial-to-mesenchymal transition; mBC: metastatic breast cancer; Pts: patients.

**Table 3 cancers-12-03588-t003:** The potential treatment response markers expressed on CTCs for future precision medicine.

Cancer Type	Molecular Marker on CTC	Detection Method	Clinical Significance as Biomarker	Positivity of Total Pts	Ref.
mCRPC	Nuclear-localized AR-V7	Immunofluorescent staining	Therapeutic response of taxane	22%	[176]
Metastatic melanoma	PD-L1	Multiparametric flow cytometry	Therapeutic response of pembrolizumab and longer PFS	64%	[174]
SCLC	DLL3+/CD45	CellSearch^®^	Therapeutic response of etoposide/platinum and decreased PFS	74.10%	[177]
Rectal cancer	TYMS/RAD23B	ISET^®^	Predict resistance of neoadjuvant chemoradiotherapy	100% (TYMS mRNA)/75% (RAD protein)	[178]
NSCLC	PD-L1	CellSearch^®^	Resistance to PD-1/PD-L1 blockade therapies	47%	[170]
Metastatic thyroid cancer	EpCAM-/CD45-/DAPI+/CEP8	Aneuploidy	Poor response to 131I treatment and worse prognosis	86.11%	[179]
Advanced gastrointestinal tumor	PD-L1	Immunofluorescent staining	Therapeutic response of PD-1/PD-L1 blockade therapies	74%	[175]
mBC	Topoisomerase 1	ApoStream (enrich for CTCs)	Therapeutic response of topoisomerase 1 inhibitor etirinotecan pegol	52%	[151]
Advanced NSCLC	CEA/hTERT	CTC chip	Therapeutic response of nivolumab, PD-1 inhibitor	Only selected patients	[180]
Advanced NSCLC	PD-L1	ISET	Poor response to nivolumab, PD-1 inhibitor and shorter PFS	83%	[172]
mCRPC	Synaptophysin	CellSearch^®^	Resistance to AR-targeted therapies, abiraterone, and enzalutamide	Only selected patients	[150]
SCLC	Bcl-2+	Immunofluorescent staining	Prognostic and treatment efficacy	72.70%	[152]
mCRPC	AR splice variants	CellSearch^®^	Therapeutic response of taxane	67%: AR-V7+, 78%: ARv567es+	[161]
mCRPC	AR splice variants	Flow cytometry	Therapeutic response of AR-targeted therapies, abiraterone, and enzalutamide	26.5%: CTC cluster(+)/AR-V7(+)	[159]
mCRPC	AR nuclear localization (%ARNL)	CellSearch^®^	Suggested resistance to taxane with increasing PSA	Only selected patients	[162]
Advanced NSCLC with ALK rearrangement	ALK-CNG	ISET	Therapeutic response of ALK inhibitor, crizotinib, by decreasing CTC of ALK-CNG	100%	[156]
mCRPC	AR splice variants	EpCAM-based Prostate Cancer Select kit	Therapeutic response of AR-targeted therapies, abiraterone. and enzalutamide	17.8%: AR-V7+	[160]
mBC	PIK3CA mutations	CellSearch^®^	Resistance to anti-HER2 therapy	36.40%	[181]
mBC	HER2	CellSearch^®^	Therapeutic response of anti-HER2 therapy	37.9	[147]
mCRPC	Nuclear-localized AR-V7	Immunofluorescent staining	Resistance to AR inhibitors	18%	[182]
Advanced NSCLC	PD-L1	CellSearch^®^	Resistance to nivolumab, PD-1 inhibitor	93%	[183]

mCRPC: metastatic castration-resistant prostate cancer; NSCLC: non-small cell lung cancer; SCLC: small cell lung cancer; mBC: metastatic breast cancer; AR: androgen receptor; ALK-CNG: ALK-copy number gain; PFS: progression-free survival; TYMS: thymidylate synthase; RAD23B: UV excision repair protein RAD23 homolog B; Pts: patients.

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
