# Peer review of "Mesenchymal Characteristics and Predictive Biomarkers on Circulating Tumor Cells for Therapeutic Strategy"

_cancers, 2020, doi:10.3390/cancers12123588_

Round 1

Reviewer 1 Report

Authors aimed to present current opinions regarding the clinical significance of CTCs and CTM with a mesenchymal phenotype as clinical surrogate markers, and to summarize the therapeutic strategy according to phenotype characterization of CTCs in various types of cancers for future precision medicine. Article is concisely written; conclusions are supported by the data; however, some minor issues should be resolved. 

Major points -        none  

Minor points:

-        I suggest to add more data regarding prognostic value of CTC with EMT phenotype in breast cancer (e.g.Anticancer Res. 2019 Apr;39(4):1829-1837, J Cancer. 2012;3:369-80) as well as experimental data regarding limitation of assays utilizing enrichment based on epithelial markers like EpCAM, cooperation between epithelial and mesenchymal cell in metastatic cascade (e.g. Cancer Res. 2006, 66, 11271–11278, Cancer Res.2009, 69, 7135–7139)-        I suggest to ongoing trial utilizing CTC with EMT phenotype in treatment decision making

Author Response

Response to the Comments of Reviewer 1

Dear reviewer,

We thank you for your time and effort in reviewing our manuscript. We are grateful for your insightful comments and suggestions, from which our manuscript has benefited enormously. We have made the necessary changes following your suggestions, and the corresponding changes made in the main manuscript are highlighted. Below, we have prepared a point-point response to all your comments.

We believe the incorporated changes as suggested have satisfactorily addressed your raised concerns. Thank you for your consideration.

Sincerely,

Shinsaku Togo

Department of Respiratory Medicine,

Juntendo University, School of Medicine

2-1-1 Hongo, Bunkyo-ku,

Tokyo 113-8421, Japan

Tel: (+81)3-5802-1063

Fax: (+81)3-5802-1617 

Point 1: I suggest to add more data regarding prognostic value of CTC with EMT phenotype in breast cancer (e.g.Anticancer Res. 2019 Apr;39(4):1829-1837, J Cancer. 2012;3:369-80) 

Response 1: Thank you for your suggestion. We have added the following text and references regarding the prognostic value of EMT-CTCs in breast cancer.

Line 134–137; page 4,

“In a study conducted for primary breast cancer, 77 (18.0%) of 427 primary breast cancer patients were found to have EMT-CTCs characterized by TWIST, SNAIL1, SLUG and ZEB1 expression level, and patients without EMT-CTCs had a longer disease-free survival than those with EMT-CTCs [53].”

Line 140–143; page 4,

“A study of metastatic breast cancer has demonstrated that overexpression of EMT-inducing transcription factors is associated with shorter PFS, and EMT-CTCs are associated with recurrence [55].”

Point 2: (following the above) as well as experimental data regarding limitation of assays utilizing enrichment based on epithelial markers like EpCAM,

Response 2: Thank you for your valuable suggestion. We have added the following paragraph and reference describing the limitations of the epithelial marker-based CTC detection methods. Please refer to lines 75–77; page 2.

 “Similarly, a study conducted in malignant pleural mesothelioma (MPM), a highly aggressive malignancy using the CellSearch® system, showed a low detection rate of 33%, suggesting a need for EpCAM-independent detection systems of CTCs [24].”

Point 3: (following the above) cooperation between epithelial and mesenchymal cell in metastatic cascade (e.g. Cancer Res. 2006, 66, 11271–11278, Cancer Res.2009, 69, 7135–7139)

Response 3: We appreciate your suggestion. We have added the suggested reference (citation number 48 Cancer Res. 2006, 66, 11271–11278) and updated the text accordingly. Please refer to lines 123 on page 4.

“… extravasation into secondary metastatic sites [48] reportedly induces the re-expression of ...”.

In addition, we have also added the following paragraph and reference describing the cooperation of epithelial and mesenchymal cells in metastasis.

Line 204–207; page 7,

“A model for cancer metastasis has been proposed in which EMT promotes the release of both EMT and non-EMT cells from the primary tumor and vascular invasion, and non-EMT cancer cells in the bloodstream collaborate with EMT cells to establish colonies in the secondary sites [106].”

Point 4:  I suggest to ongoing trial utilizing CTC with EMT phenotype in treatment decision making.

Response 4: We agree with your specific suggestion. We have added a new section on EMT and predictive biomarker analysis for treatment decision-making with the heading “5.4 The combination analysis of EMT and predictive markers for treatment decision making.”

Please refer to lines 351–375; pages 11–12. The revised text reads;

“5.4. The combination analysis of EMT and predictive markers for treatment decision making

The existence of EMT-CTC was found to correlate with drug resistance [54,185]. Therefore, the detection and monitoring of EMT as a marker of treatment resistance before and during treatment is important to improve the quality of treatment management. High EMT marker expression has also been associated with high expression of immune checkpoint markers, such as PD1, PD-L1, CTLA4, OX40L, and PD-L2 [186-189]. For example, several studies have shown that EMT in cancer cells increases the expression of PD-L1 and consequently confers immune tolerance in cancer cells [72,190-193]. Analyses of CTCs in NSCLC patients revealed that PD-L1 expression on EMT-CTCs associated with significantly poorer survival after curative surgery, and PD-L1 and EMT markers were expressed at significantly higher proportions in CTCs than patient-matched NSCLC tissues [72,191]. Furthermore, it was reported that PD-L1 expressed in the nucleus of vimentin-positive CTCs predicts poor prognosis in colorectal and prostate cancer patients [194]. Several studies suggest complex bidirectional regulation between EMT and PD-L1 signaling, leading to tumor immune escape [193-195]. The combined analysis of EMT and PD-L1 expression in CTCs could be an effective method for predicting and monitoring the outcome of immunotherapy. EMT-related genes in CTCs are also candidate markers for combination analysis. For example, in metastatic prostate cancer, IGF1, IGF2, EGFR, FOXP3, and TGFB3, were commonly observed in CTCs of castration-sensitive cases, whereas an additional subset of EMT-related genes (e.g., PTPRN2, ALDH1, ESR2, and WNT5A) were expressed in CTCs of castration-resistant cases [198]. Mesenchymal phenotype and increased expression level of AR-V7 in CTCs from patients with mCRPC has been observed for those with a limited response to abiraterone and enzalutamide (androgen-signaling targeted inhibitors) treatment [164]. Further detailed molecular analysis of the relationship between EMT and predictive biomarkers could lead to a better understanding of the mechanisms of treatment resistance and create a reliable basis for therapeutic decision making.”

Reviewer 2 Report

In the manuscript »Circulating tumor cells with mesenchymal phenotypes as clinical surrogate biomarkers and therapeutic strategy« the authors present an interesting review of the field of translational CTC research. In the introduction section of the article they describe well the role of EMT in tumor progression and the difference between epithelial and mesenchymal phenotype of CTC. They point out the importance of mesenchymal traits of CTC for the progression of the disease and thus the prognosis of patients. They introduce the mesenchymal markers for different tumor types. Additionally, they introduce the stem cell markers within the mesenchymal type of CTC. At the end they show the concordance between the primary tumor and CTC mutations and their role as potential treatment response markers.

I find the review interesting, fair and relevant. However, I somehow miss the mentioning of the greatest drawback of CTC detection - namely their rare occurrence in the peripheral blood of (even metastatic) patients. That would shed some more light on the potential clinical usefulness of the further characterization of CTC within the concept of liquid biopsy. It is on my opinion also the reason, why the translational research is focusing much more on the ctDNA than CTC lately.

Apart from that comment I have no further objections.

Author Response

Response to the comments of Reviewer 2

Dear reviewer,

We thank you for your time and effort in reviewing our manuscript. We are grateful for your insightful comments and suggestions, from which our manuscript has benefited enormously. We have made the necessary changes following your suggestions, and the corresponding changes made in the main manuscript are highlighted. Below, we have prepared a point-point response to all your comments.

We believe the incorporated changes as suggested have satisfactorily addressed your raised concerns. Thank you for your consideration.

Sincerely,

Shinsaku Togo

Department of Respiratory Medicine,

Juntendo University, School of Medicine

2-1-1 Hongo, Bunkyo-ku,

Tokyo 113-8421, Japan

Tel: (+81)3-5802-1063

Fax: (+81)3-5802-1617

Point 1: I find the review interesting, fair and relevant. However, I somehow miss the mentioning of the greatest drawback of CTC detection - namely their rare occurrence in the peripheral blood of (even metastatic) patients. That would shed some more light on the potential clinical usefulness of the further characterization of CTC within the concept of liquid biopsy.

Response 1: Thank you for all praises. We agree with your opinion. The detection of CTCs based on the epithelial markers has a low sensitivity. Moreover, the detection rate of CTCs is different because of the diverse CTCs detection method, as shown in Table 1. Therefore, it is important to develop a strategy/ system, which will enable CTC detection with high-sensitivity. EMT occurs from the early phase of carcinogenesis; therefore, detecting EMT-CTCs from the early stage of cancer to even metastatic advanced stage could result in an increased sensitivity of CTCs detection than the epithelial marker-based detection such as CellSearch system. We have described this in section 2.1. However, to aid clarity, we have added a new concluding paragraph to section 2.1 (Lines 187–190; page 6), and the revised text reads;

“Thus, the development of a system with EMT-CTCs detection capability will enable high-sensitivity CTC detection from early stages of cancer to advanced metastatic stages compared to systems that use only epithelial markers to detect CTCs such as CellSearch® system.”

Point 2: It is on my opinion also the reason, why the translational research is focusing much more on the ctDNA than CTC lately.

Response 2: Thank you for raising an important concern. We agree with your opinion. ctDNA analysis has been widely used to monitor the response to therapy; its detection is tricky. Studies comparing the effects of CTC and ctDNA have shown an association between the CTC numbers and the OS of patients, while the prognostic impact of baseline ctDNA level could not be detected. On the contrary, recent studies show that these two competing biomarkers could be complementary: ctDNA provides a cost-effective and highly sensitive tool for the detection of mutations, in particular in the selection of cancer-associated hotspots, while CTCs represent a useful tool to investigate drug sensitivity, tumor aggressiveness and heterogeneity, mutations at a genome-wide scale, and also RNA and protein expression.  (Fici P. Cell-Free DNA in the Liquid Biopsy Context: Role and Differences Between ctDNA and CTC Marker in Cancer Management. Methods Mol Biol. 2019;1909:47-73. DOI: 10.1007/978-1-4939-8973-7_4. PMID: 30580422, and reference thereof).

In our manuscript, though not in detail, we have described the usefulness of CTCs over ctDNA in the introduction section (line numbers 56–68; page 2). However, there was a typographical error (we had mistakenly written ‘ctDNA’ as ‘cfDNA’), which we think might have affected its understanding. We have corrected the error, and the revised text reads;

“Therefore, the significance of CTCs as clinical surrogate markers is different from that of other diverse liquid biopsies (i.e., circulating free DNA, ctDNA, exosomes, or diverse serum liquid biomarkers) [16-19]. Among these, ctDNA measurement is the most popular diagnostic and prognostic tool that averages the status of the whole tumor burden and provides information on the response/resistance to chemotherapy, targeted therapy, and immunotherapy; therefore, it is useful for decision-making [17-18]. However, these biomarkers are considered as indirect markers released from tumor tissues into the bloodstream. In contrast, detection of CTCs and circulating tumor microemboli (CTM) provide direct evidence for the existence of cancer lesions, and thus their measurement and phenotypic characterization could be a reliable diagnostic tool for cancer.”

Reviewer 3 Report

The title of the proposed review is very attractive as targeting EMT to potentially cure cancer is still an unmet scientific/clinical goal. Overall, the review is nicely written and comprehensive.

A major issue, however, is that the content of the review does not really match the title as there is very little description of the link between the markers targeted on CTC and EMT. The authors should therefore implement more scientific background on the potential links between the markers studied in CTC (chapter 6) and EMT, as they have done lines 238 or 257 only. Note that the statement lines 280-282 should therefore belong to chapter 6.3. The work of JP Thiery could be helpful in this regard for example. also, is there a particular link of ARV7 and mesenchymal phenotype already described?

Another missing point would be a paragraph describing the challenges to detect EMT in CTC and making sure that the cells with such mesenchymal phenotype are tumoral (due to the usage of mesenchymal markers also present in leukocytes).

I miss also the link between the chapter 4 and the chapter 5. Maybe chapter 5 should be included as an introdcution of chapter 6. In any case, the title needs to be changed cause it is not only the phenotype that is discussed but also the genotype.

There are several points that need to be adressed:

  • some statements in the introduction are imprecise: for example, the authors should precise in which clinical entities CTCs are used as surrogate endpoints markers (line 34).  and also remind more generally that they have pronostic value in several cancer entities (and quote the appropriate references, line 36)
  • ctDNA has also a strong pronostic value, therefore the statements line 48_50 need to be revised. and it can provide information on resistance to other drugs than chemotherapy, like targeted therapy or immunotherapy.
  • line 69: new papers from Aceto should be mentionned.
  • line 100: precise the difference between cell surface vimentin and vimentin, and what this difference implies.
  • line 115: quote some references to justify this statement
  • line 118-120: better explicite how EMT-CTC are dispensable, does it mean that these cells have no pronostic value?
  • chapter 2.1: there are very strong data on CTC and early stages of breast cancer to quote as well at the beginning of the chapter. maybe would be worth checking if there are any data on EMT in this setting as well?
  • line 262: define CSV+

Author Response

Response to the comments of Reviewer 3 

Dear reviewer,

We thank you for your time and effort in reviewing our manuscript. We are grateful for your insightful comments and suggestions, from which our manuscript has benefited enormously. We have made the necessary changes following your suggestions, and the corresponding changes made in the main manuscript are highlighted. Below, we have prepared a point-point response to all your comments.

We believe the incorporated changes as suggested have satisfactorily addressed your raised concerns. Thank you for your consideration.

Sincerely,

Shinsaku Togo

Department of Respiratory Medicine,

Juntendo University, School of Medicine

2-1-1 Hongo, Bunkyo-ku,

Tokyo 113-8421, Japan

Tel: (+81)3-5802-1063

Fax: (+81)3-5802-1617

Point 1: A major issue, however, is that the content of the review does not really match the title as there is very little description of the link between the markers targeted on CTC and EMT. The authors should therefore implement more scientific background on the potential links between the markers studied in CTC (chapter 6) and EMT, as they have done lines 238 or 257 only.

Response 1:  Thank you for your valuable suggestion. We agree with your opinion and accordingly aid more scientific background on the potential links between the markers studied in CTC and EMT. We have added a new section, “5.4 The combination analysis of EMT and predictive markers for treatment decision making.” Please refer to lines 351–375 on pages 11–12. The revised text reads;

5.4. The combination analysis of EMT and predictive markers for treatment decision making

The existence of EMT-CTC was found to correlate with drug resistance [54,185]. Therefore, the detection and monitoring of EMT as a marker of treatment resistance before and during treatment is important to improve the quality of treatment management. High EMT marker expression has also been associated with high expression of immune checkpoint markers, such as PD1, PD-L1, CTLA4, OX40L, and PD-L2 [186-189]. For example, several studies have shown that EMT in cancer cells increases the expression of PD-L1 and consequently confers immune tolerance in cancer cells [72,190-193]. Analyses of CTCs in NSCLC patients revealed that PD-L1 expression on EMT-CTCs associated with significantly poorer survival after curative surgery, and PD-L1 and EMT markers were expressed at significantly higher proportions in CTCs than patient-matched NSCLC tissues [72,191]. Furthermore, it was reported that PD-L1 expressed in the nucleus of vimentin-positive CTCs predicts poor prognosis in colorectal and prostate cancer patients [194]. Several studies suggest complex bidirectional regulation between EMT and PD-L1 signaling, leading to tumor immune escape [193-195]. The combined analysis of EMT and PD-L1 expression in CTCs could be an effective method for predicting and monitoring the outcome of immunotherapy. EMT-related genes in CTCs are also candidate markers for combination analysis. For example, in metastatic prostate cancer, IGF1, IGF2, EGFR, FOXP3, and TGFB3, were commonly observed in CTCs of castration-sensitive cases, whereas an additional subset of EMT-related genes (e.g., PTPRN2, ALDH1, ESR2, and WNT5A) were expressed in CTCs of castration-resistant cases [198]. Mesenchymal phenotype and increased expression level of AR-V7 in CTCs from patients with mCRPC has been observed for those with a limited response to abiraterone and enzalutamide (androgen-signaling targeted inhibitors) treatment [164]. Further detailed molecular analysis of the relationship between EMT and predictive biomarkers could lead to a better understanding of the mechanisms of treatment resistance and create a reliable basis for therapeutic decision making.”

Furthermore, we have revised the title of the article to emphasize the relationship between the markers studied in CTC and EMT. The revised title reads;

“Mesenchymal characteristics and predictive biomarkers on circulating tumor cells for therapeutic strategy.”

Point 2: Note that the statement lines 280-282 should therefore belong to chapter 6.3.

Response 2: As suggested, we have moved the indicated sentence to lines 354–356 under section 5.4 on page 11.

Point 3: The work of JP Thiery could be helpful in this regard for example. also, is there a particular link of ARV7 and mesenchymal phenotype already described?

Response 3:  Thank you for suggesting to add the findings of Terry et al. (citation number 197). It has provided added evidence for the relation between CTC markers and EMT.  The text citing this reference reads;

“Several studies suggest complex bidirectional regulation between EMT and PD-L1 signaling, leading to tumor immune escape [195-197].” Lines 363–364; Page 11.

Furthermore, to describe the particular link of ARV7 and mesenchymal phenotype, we have added a new reference (number 164) and updated the text accordingly. The added texts are incorporated in lines 326–328 (page 9) and 370–372 (page 12)and read as follows;

“Lines 326–328 (page 9): Phenotypic profiling of CTCs isolated from patients with metastatic CRPC (mCRPC) has revealed elevated levels of AR-V7 and the mesenchymal marker, N-cadherin, in patients with progressive cancer [164].

Lines 370–372 (page 12): Mesenchymal phenotype and increased expression level of AR-V7 in CTCs from patients with mCRPC has been observed for those with a limited response to abiraterone and enzalutamide (androgen-signaling targeted inhibitors) treatment [164].”

Point 4: Another missing point would be a paragraph describing the challenges to detect EMT in CTC and making sure that the cells with such mesenchymal phenotype are tumoral (due to the usage of mesenchymal markers also present in leukocytes).

Response 4: Thank you for the suggestion. We have added the following texts and references. Please refer to lines 161–167, page 6.

 “Since mesenchymal markers such as N-Cadherin and vimentin are frequently expressed in blood cells, it is necessary to remove these blood cells during the enrichment and detection processes of CTCs. One approach to enable blood cell removal is the negative selection with blood cell-specific markers, such as CD45, not expressed in cancer cells. Although there are no universal markers for specific enrichment and detection of mesenchymal CTCs, various detection systems have combined negative selection of blood cells with positive selection by mesenchymal markers to achieve highly sensitive EMT-CTCs detection [85-89].”

Point 5: I miss also the link between the chapter 4 and the chapter 5. Maybe chapter 5 should be included as an introdcution of chapter 6. In any case, the title needs to be changed cause it is not only the phenotype that is discussed but also the genotype.

Response 5: Apologies for the confusion. However, we wanted to discuss the discrepancies between the characteristics of the primary site and CTCs; therefore, to aid clarity, we have added the following paragraph (lines 276–283) instead of merging section 5 and section 6.

“These discordances could be, in part, due to the inability of traditional biopsies that sample a small piece of tumor tissue to reflect the heterogeneity of cancer cells in tumor tissue leading to failure in cancer prognosis and prediction. In contrast, several studies have provided evidence that the heterogeneity of CTCs reflects the cellular heterogeneity of the primary and metastatic tumor [141-143]; moreover, CTCs often acquire different characteristics from their original tissue during the metastatic process. Therefore, a detailed characterization of the expression of CTC markers could provide useful information in predicting prognosis and treatment response as a real-time biopsy [140,144].”

 Furthermore, we have revised the heading of section 4 (it was section 5 in our earlier submission) to “The concordance between the characteristics of CTCs and primary tumors.”

Point 6: some statements in the introduction are imprecise: for example, the authors should precise in which clinical entities CTCs are used as surrogate endpoints markers (line 34).  and also remind more generally that they have pronostic value in several cancer entities (and quote the appropriate references, line 36)

Response 6: We agree with your suggestions. We have replaced the original sentence with the following texts and references to clarify in which cancer types, CTCs are used as surrogate markers and described CTC’s generality as a prognostic marker.

Lines 44–47; page 2;

“Increasing numbers of circulating tumor cells (CTCs) are now utilized as surrogate endpoint markers [1-5], and many clinical studies have demonstrated that the presence of CTCs in the blood of patients associates with short recurrence-free survival and high risk of metastasis for several types of solid cancers [6-13].”  

Point 7: ctDNA has also a strong pronostic value, therefore the statements line 48_50 need to be revised. and it can provide information on resistance to other drugs than chemotherapy, like targeted therapy or immunotherapy.

Response 7: We agree with your comments. We have revised the texts and references accordingly, describing the prognostic value and information on response/resistance to targeted therapy and immunotherapy of ctDNA, as follows;

Lines 58–61; page 2

“Among these, ctDNA measurement is the most popular diagnostic and prognostic tool that averages the status of the whole tumor burden and provides information on the response/resistance to chemotherapy, targeted therapy, and immunotherapy; therefore, it is useful for decision-making [17-18].”

Point 8: line 69: new papers from Aceto should be mentionned.

Response 8: Thank you for your suggestion. We have added the suggested reference (citation number 34) as supporting evidence to the text. Please refer to lines 83–84; page 2.

[Aceto N. Bring along your friends: Homotypic and heterotypic circulating tumor cell clustering to accelerate metastasis. Biomed J. 2020;43(1):18-23. doi:10.1016/j.bj.2019.11.002]

Point 9: line 100: precise the difference between cell surface vimentin and vimentin, and what this difference implies.

Response 9: We have added a paragraph describing the differences between vimentin and cell-surface vimentin (CSV).

Line 117–121

“Vimentin, a cytoplasmic protein comprising intermediate filaments, is known to be up-regulated in EMT-induced cancer cells and correlated with cancer progression [39-41]. In addition, some vimentin molecules expressed in cancer cells migrate to the cell surface and have been identified as cancer-specific EMT markers, called cell-surface vimentin (CSV), and were shown to correlate with highly invasive and metastatic properties of cancers [42-47].”

Point10 : line 115: quote some references to justify this statement

Response 10: Thank you for your suggestion. We have added the following three references to justify the statement, “The presence of mesenchymal markers, Twist1+, vimentin+, or CSV+, in CTCs predict a worse prognosis than the expression of epithelial markers alone in metastatic cancers [46,47,54]”. Please see lines 138–140; page 4.

  1. Liu, M.; Wang, R.; Sun, X.; Liu, Y.; Wang, Z.; Yan, J.; Kong, X.; Liang, S.; Liu, Q.; Zhao, T., et al. Prognostic significance of PD-L1 expression on cell-surface vimentin-positive circulating tumor cells in gastric cancer patients. Mol Oncol 2020, 14, 865-881, doi:10.1002/1878-0261.12643.
  2. Satelli, A.; Batth, I.; Brownlee, Z.; Mitra, A.; Zhou, S.; Noh, H.; Rojas, C.R.; Li, H.; Meng, Q.H.; Li, S. EMT circulating tumor cells detected by cell-surface vimentin are associated with prostate cancer progression. Oncotarget 2017, 8, 49329-49337, doi:10.18632/oncotarget.17632.
  3. Agnoletto, C.; Corra, F.; Minotti, L.; Baldassari, F.; Crudele, F.; Cook, W.J.J.; Di Leva, G.; d'Adamo, A.P.; Gasparini, P.; Volinia, S. Heterogeneity in Circulating Tumor Cells: The Relevance of the Stem-Cell Subset. Cancers (Basel) 2019, 11, doi:10.3390/cancers11040483

Point11 : line 118-120: better explicite how EMT-CTC are dispensable, does it mean that these cells have no pronostic value?

Response 11:  Thank you for your suggestion. We have revised the text by adding the following paragraph.

“These negative results for the value of EMT-CTCs as prognostic markers are probably due to the diverse mechanisms that induce EMT and the lack of ability to detect EMT-CTC subtypes in clinical settings [67].” Lines 150–152; page 5.

Point 12: chapter 2.1: there are very strong data on CTC and early stages of breast cancer to quote as well at the beginning of the chapter. maybe would be worth checking if there are any data on EMT in this setting as well?

Response 12: We agree with you that several studies have reported the applicability of CTCs in breast cancer. We have added the following text and references to describe the relation between EMT-CTC and early breast cancer.

“A large meta-analysis of 19 studies enrolling 2993 early-stage breast cancer patients demonstrated the detection of the CTCs as a stable prognosticator in early-stage breast cancer [90]. Several studies have identified CTCs expressing EMT markers in patients with early-stage breast cancer [57,91,92] at a higher frequency than those expressing the epithelial markers [91]. Furthermore, RT-PCR characterization of CTCs isolated from early-stage breast cancer patients, where TWIST1 expression was used as an EMT marker, revealed the presence of EMT-CTCs in 31% of patients [93], whereas EMT-CTC markers expressing Twist1(+) and vimentin(+), were identified in 73% and 77% of patients with early-stage breast cancer, respectively [57].” Lines 169–177; page 6.

Point13 : line 262: define CSV+

Response 13:  Thank you for your suggestion. We have defined CSV to line 119-121. Please refer to the response to your comment number 9.

Reviewer 4 Report

In their current manuscript, Okabe et al. summarize our current knowledge on circulating tumor cells as potential biomarkers and focusing particularly on their epithelial/mesenchymal characteristics, the concordance of their phenotype with primary tumors as well as a presence of molecular markers that could facilitate disease prognosis and personalized treatment decisions. The review is reasonably structured, rather well written and covers in a satisfactory manner the literature on these topics.

Specific comments:

  1. The title reads ‘ Circulating tumor cells … as therapeutic strategy’. As CTCs are not a ‘therapeutic strategy’, this part of the title should be revised.
  2. Legend title for Figure 1 (‘Schematic representation of CTCs and metastatic process’) does not correspond well to the actual figure as it shows rather the way(s) how CTCs are isolated and characterized for the purpose of clinical use rather than showing the metastatic process (only).
  3. Page 4, line 121 – ‘show KRAS mutation in CTCs’ – do these patients show one particular KRAS mutation? If so, please, state which one, otherwise should be ‘show KRAS mutations’.
  4. There are inconsistencies in chapter numbering – chapters 3, 7, 8 and 9 are actually ‘missing’.
  5. Chapter 5 on the concordance of CTCs with primary tumors is somehow short and it would be of interest, if more studies (if available) would be mentioned and discussed. Also, are there any plausible explanations/hypotheses in the literature that would clarify why for some molecular markers (e.g., KRAS) a concordance is reported in particular tumor types whereas for other (e.g., HER2) not? It would be interesting if the authors could elaborate on this issue within this chapter as well.
  6. Page 8, line 215 – ‘trastuzumab…did not respond to…’ – this needs to be reformulated as trastuzumab is a targeting agent and is not expected to respond; tumors do or don’t respond TO targeting agents.
  7. Page 8, line 229-230 – ‘The efficacy and prognosis…of TKI treatment, the molecular target of EFGR driver mutations…’ – again, this is a linguistic nonsense as TKI treatment is NOT a target; EGFR (or EGFR driver mutation)s ARE TARGETS of TKIs; please, reformulate;
  8. Page 8, line 249-251 – ‘Programmed….of melanoma and NSCLC.’ – please, add a reference (or references) for this statement.
  9. Page 10, line 274-275 – ‘Therefore, the expression…are not necessarily the same.’ – not clear at all what is the meaning of this statement; please, reformulate the sentence;

In the discussion, ref. 123 is included. It would be useful to elaborate on this study in a more detailed manner already in chapter 6.3.

Author Response

Response to the comments of Reviewer 4

Dear reviewer,

We thank you for your time and effort in reviewing our manuscript. We are grateful for your insightful comments and suggestions, from which our manuscript has benefited enormously. We have made the necessary changes following your suggestions, and the corresponding changes made in the main manuscript are highlighted. Below, we have prepared a point-point response to all your comments.

We believe the incorporated changes as suggested have satisfactorily addressed your raised concerns. Thank you for your consideration.

Sincerely,

Shinsaku Togo

Department of Respiratory Medicine,

Juntendo University, School of Medicine

2-1-1 Hongo, Bunkyo-ku,

Tokyo 113-8421, Japan

Tel: (+81)3-5802-1063

Fax: (+81)3-5802-1617

Point 1: The title reads ‘Circulating tumor cells … as therapeutic strategy’. As CTCs are not a ‘therapeutic strategy’, this part of the title should be revised.

Response 1: Thank you for your suggestion. We have changed the title to “Mesenchymal characteristics and predictive biomarkers on circulating tumor cells for therapeutic strategy.”

Point 2 : Legend title for Figure 1 (‘Schematic representation of CTCs and metastatic process’) does not correspond well to the actual figure as it shows rather the way(s) how CTCs are isolated and characterized for the purpose of clinical use rather than showing the metastatic process (only).

Response 2: Thank you for highlighting the mistake. We have changes the figure caption as follows,

Schematic representation of circulating tumor cells (CTCs) during the metastatic process and characterization as a clinical biomarker after isolation.” Lines 87–88; page 3.

Point 3: Page 4, line 121 – ‘show KRAS mutation in CTCs’ – do these patients show one particular KRAS mutation? If so, please, state which one, otherwise should be ‘show KRAS mutations’.

Response 3: The cited paper has detected various KRAS mutations in codons 12 and 13 from CTCs by PCR. Therefore, we have changed the texts as follows.

“Patients with PDAC show KRAS mutations in CTCs [64,65], indicating a prognostic value, which does not depend on the expression of ZEB1, the EMT marker [66].” Lines 148–152; page 5.

Point 4 : There are inconsistencies in chapter numbering – chapters 3, 7, 8 and 9 are actually ‘missing’.

Response 4: Thank you. We have updated the section numbers.

Point5 Chapter 5 on the concordance of CTCs with primary tumors is somehow short and it would be of interest, if more studies (if available) would be mentioned and discussed. Also, are there any plausible explanations/hypotheses in the literature that would clarify why for some molecular markers (e.g., KRAS) a concordance is reported in particular tumor types whereas for other (e.g., HER2) not? It would be interesting if the authors could elaborate on this issue within this chapter as well.

Response 5: Thank you for your insightful suggestion.

We have revised the text by describing the concordance and discordance in KRAS and HER2 expression between CTCs and primary tumors and the factors that may have caused the discordances. Finally, we concluded that the detailed characterization of CTC as a real-time biopsy is important for predicting prognosis and treatment response, as follows.

Lines 254–262; page 8,;

“Buim et al. also showed that KRAS mutations (codon 12 and 13) in CTCs and matched primary tumor from patients with colorectal cancer showed a high grade of concordance of 71% (P = 0.017), which suggests that CTCs can be used as surrogates of primary tumors in clinical practice [126]. Several other studies have also shown similar results [126-129]. Kalikaki et al. characterized CTCs from 31 metastatic colorectal cancer patients for KRAS mutations on codons 12 and 13 and showed 45% and 16.7% of patients with mutant and wild-type primary tumors, respectively have detectable mutations in their CTCs [130]. They further revealed that individual patients’ CTCs in serial blood samples exhibited different mutational statuses of KRAS during treatment [130].”

Lines 267–272; page 8;

“In breast cancer, the human epidermal growth factor receptor 2 (HER2) oncogene has been shown to play an important role in the growth and progression [131,132]. A good correspondence of HER2 amplification was found between CTCs and the primary tumor, as well as the immunohistochemistry results [133], suggesting that the status of HER2 on CTC reflects the status of the primary tumor. However, several studies have demonstrated that HER2 status might change during disease progression [134-137].”

Lines 276–283; page 9;

“These discordances could be, in part, due to the inability of traditional biopsies that sample a small piece of tumor tissue to reflect the heterogeneity of cancer cells in tumor tissue leading to failure in cancer prognosis and prediction. In contrast, several studies have provided evidence that the heterogeneity of CTCs reflects the cellular heterogeneity of the primary and metastatic tumor [141-143]; moreover, CTCs often acquire different characteristics from their original tissue during the metastatic process. Therefore, a detailed characterization of the expression of CTC markers could provide useful information in predicting prognosis and treatment response as a real-time biopsy [140,144].”

Point 6 : Page 8, line 215 – ‘trastuzumab…did not respond to…’ – this needs to be reformulated as trastuzumab is a targeting agent and is not expected to respond; tumors do or don’t respond TO targeting agents.

Response 6: Thank you for highlighting the mistake. We have rewritten it as follows;

Lines 294-296; page 9;

“However, trastuzumab, an anti-HER2 antibody therapy, did not decrease the detection rate of CTCs in patients with non-HER2-amplified early breast cancer following surgery after chemotherapy [149].”

Point 7 : Page 8, line 229-230 – ‘The efficacy and prognosis…of TKI treatment, the molecular target of EFGR driver mutations…’ – again, this is a linguistic nonsense as TKI treatment is NOT a target; EGFR (or EGFR driver mutation)s ARE TARGETS of TKIs; please, reformulate;

Response 7: Apologies for the language error. We have modified the text as follows;

Lines 308–310; page 9,

“The efficacy and prognosis of EGFR-tyrosine kinase inhibitor (TKI) treatment for the patients with EGFR driver mutations were reportedly better in the low CTC group than in the high CTC group in patients with NSCLC [154].”

Point 8: Page 8, line 249-251 – ‘Programmed….of melanoma and NSCLC.’ – please add a reference (or references) for this statement.

Response 8: Thank you for the suggestion. We have added the references and modified the text as follows;

Lines 330–332; page 10;

“Programmed death-ligand 1 (PD-L1) is widely implicated in tumor immune evasion, and PD-L1 expression in tumor tissues has been reported to be associated with the responses to the current checkpoint immunotherapy of melanoma and NSCLC [165-169].”

Point 9: Page 10, line 274-275 – ‘Therefore, the expression…are not necessarily the same.’ – not clear at all what is the meaning of this statement; please, reformulate the sentence;

Response 9: Apologies for our oversight. We have rephrased the sentence as follows;

“… the CTCs and CTM with a mesenchymal phenotype are associated with poor clinical outcomes and provide information on recurrence periods, rendering them as a potential source of predictable biomarkers of chemoresistance. Therefore, detailed analyses of multivariate targets in CTCs could improve clinical decision making as a real-time liquid biopsy.” Lines 383–389; page 12.

Point 10: In the discussion, ref. 123 is included. It would be useful to elaborate on this study in a more detailed manner already in chapter 6.3.

Response 10: Thank you for your suggestion. We agree with your opinion. We have added a new section titled “5.4. The combination analysis of EMT and predictive markers for treatment decision making.” on page 11, discussing the relationship between EMT and clinical predictive biomarkers.

We also have revised the conclusion to highlight the future implications and new research directions of this study.

Lines 389-392; page 12;,

“This study suggests that for the development of specific, reliable therapeutic markers based on CTCs and CTM, further clinical prospective studies should be undertaken to increase the clinical value of CTC and CTM measurements for future personalized treatments.”